# Pyrotinib plus docetaxel as first-line treatment for HER2-positive metastatic breast cancer: the PANDORA phase II trial

Yabing Zheng[1,11], Wen-Ming Cao [1,11], Xiying Shao[1], Yanxia Shi[2], Li Cai[3], Wenyan Chen[4], Jian Liu[5], Peng Shen[6], Yiding Chen [7], Xian Wang[8], Huiping Li[9], Man Li[10], Zhanhong Chen[1,12] ✉ & Xiaojia Wang [1,12] ✉

The role of pyrotinib in the treatment of HER2-positive metastatic breast cancer (MBC) has been well-established. This multicenter, single-arm phase II trial (NCT03876587) aimed to assess the benefit of pyrotinib plus docetaxel as a first-line treatment for HER2-positive MBC. Women with HER2-positive MBC who had not undergone HER2 blockade or chemotherapy for metastatic disease were enrolled in the study and received daily oral pyrotinib 400 mg plus intravenous docetaxel 75 mg/m$^2$ every 3 weeks. The primary endpoint was the objective response rate (ORR), secondary endpoints included progression-free survival (PFS), duration of response (DoR), clinical benefit rate (CBR), overall survival (OS) and safety. From June 2019 to June 2021, 79 patients were enrolled. The confirmed ORR was 79.7% (95% confidence interval [CI], 70.8-88.6), and the CBR was 87.3% (95%CI, 80.0-94.6) in the intention-to-treat population. The pre-specified primary endpoint was met. The median DoR was 15.9 months (interquartile range, 8.3-19.5); the median PFS was 16.0 months (95% CI, 11.2-20.8), and the median OS was not reached. The most common grade ≥3 treatment-related adverse events observed were leukopenia (29.1%), neutropenia (27.8%), and diarrhea (21.5%). This study demonstrates that pyrotinib plus docetaxel show an acceptable safety profile and promising antitumor activity as a first-line treatment option for patients with HER2-positive MBC.

Human epidermal growth factor receptor 2 (HER2) targeted therapy combined with chemotherapy is the established first-line approach for treating HER2-positive metastatic breast cancer (MBC)[1]. Studies like H0648g and M77001 have demonstrated the superiority of trastuzumab in the first-line treatment of HER2-positive MBC[2,3]. Although several anti-HER2 tyrosine kinase inhibitors (TKIs) have been investigated as first-line treatments for HER2-positive MBC, no regimen has yet received approval[4,5]. The CLEOPATRA study has recommended the combination of pertuzumab, trastuzumab, and docetaxel as a preferred regimen for previously untreated HER2-positive MBC[6–8]. However, as trastuzumab, with or without pertuzumab, is increasingly used in the (neo)adjuvant setting, HER2-positive breast cancer patients

[1]Zhejiang Cancer Hospital, Hangzhou, Zhejiang, China. [2]Sun Yat-sen University Cancer Center, Guangzhou, Guangdong, China. [3]Harbin Medical University Cancer Hospital, Harbin, Heilongjiang, China. [4]Nanchang People's Hospital, Nanchang, Jiangxi, China. [5]Fujian Cancer Hospital, Fuzhou, Fujian, China. [6]The First Affiliated Hospital, Zhejiang University School of Medicine, Hangzhou, Zhejiang, China. [7]The Second Affiliated Hospital, Zhejiang University School of Medicine, Hangzhou, Zhejiang, China. [8]Sir Run Run Shaw Hospital, Zhejiang University School of Medicine, Hangzhou, Zhejiang, China. [9]Beijing Cancer Hospital, Beijing, China. [10]The Second Affiliated Hospital of Dalian Medical University, Dalian, Liaoning, China. [11]These authors contributed equally: Yabing Zheng, Wen-Ming Cao. [12]These authors jointly supervised this work: Zhanhong Chen, Xiaojia Wang. ✉e-mail: czred@sina.com; wxiaojia0803@163.com

who experience disease progression after (neo)adjuvant trastuzumab and/or pertuzumab require special attention. Consequently, more first-line treatment options are needed for such patients in the advanced setting. The treatment paradigm for HER2-positive MBC is continually evolving, with new developments such as antibody-drug conjugates (ADCs) like trastuzumab deruxtecan[9,10] and TKIs including tucatinib[11,12]. For example, the ongoing DESTINY-Breast09 trial is assessing the impact of trastuzumab deruxtecan, with or without trastuzumab, versus the combination of chemotherapy, trastuzumab, and pertuzumab as a first-line treatment for HER2-positive MBC (NCT04784715). Furthermore, the HER2CLIMB-05 trial is exploring the potential benefits of supplementing the regimen of chemotherapy, trastuzumab, and pertuzumab with tucatinib (NCT05132582). Nevertheless, the outcomes from these trials are yet to be reported.

Pyrotinib is a small molecule TKI that targets HER1, HER2, and HER4[13]. The phase III PHINIX trial demonstrated that combining pyrotinib with capecitabine led to a prolonged progression-free survival (PFS) compared to capecitabine alone in patients with HER2-positive MBC who had previously failed trastuzumab and taxane treatment[14]. Additionally, both the phase II trial and phase III PHOEBE trial have shown the superiority of pyrotinib plus capecitabine over lapatinib plus capecitabine for previously treated HER2-positive MBC[15,16]. Recently, the phase III PHILA study revealed that pyrotinib in combination with trastuzumab and docetaxel significantly extended PFS compared to placebo plus trastuzumab and docetaxel as a first-line treatment for HER2-positive MBC patients[17].

In this work, we assess the efficacy and safety of pyrotinib plus docetaxel as a first-line treatment for HER2-positive MBC. Our findings demonstrate that the combination of pyrotinib with docetaxel shows promising antitumor activity and offers a PFS benefit in the first-line treatment of patients with HER2-positive MBC.

## Results
### Study population
Between June 12, 2019 and June 18, 2021, a total of 79 patients were enrolled and received the study treatment. However, two patients were found to have major protocol deviations (one with brain metastasis and the other with heart failure). As a result, 79 patients were included in the intention-to-treat (ITT) population, while 77 patients were included in the per-protocol (PP) population. As of the data cutoff date on October 31, 2022, 27 patients were still receiving treatment

(Fig. 1). The median age of the patients was 52 years, ranging from 28 to 70 years. Out of all participants, 33 (41.8%) had previously received taxane, and 24 (30.4%) had received trastuzumab in the (neo)adjuvant setting (Table 1).

### Efficacy
During the first stage of the study, 24 out of 27 patients achieved confirmed objective responses, including one complete response (CR) and 23 partial responses (PR). Subsequently, the study proceeded to the second stage. Among all enrolled patients, five (6.3%) were deemed not evaluable: two due to major protocol violations, and three withdrew from the study due to adverse events (AEs) after the first dosing of the study treatment. Among the evaluable patients, three (3.8%) achieved CR, and 60 (75.9%) achieved PR. The confirmed objective response rate (ORR) was 79.7% (95% confidence interval [CI], 70.8-88.6) in the ITT population and 81.8% (95% CI, 73.2-90.4) in the PP population. Furthermore, the clinical benefit rates (CBR) were 87.3%

**Table 1 | Baseline characteristics**

| Characteristics | ITT (*n* = 79) | Biomarker set (*n* = 31) |
|---|---|---|
| Age (years), median (range) | 52 (28–70) | 50 (28–70) |
| *Hormone receptor status, n (%)* | | |
| ER and/or PgR positive | 44 (55.7) | 18 (58.1) |
| ER and PgR negative | 35 (44.3) | 13 (41.9) |
| *ECOG PS, n (%)* | | |
| 0 | 18 (22.8) | 3 (9.7) |
| 1 | 61 (77.2) | 28 (90.3) |
| *Metastasis site, n (%)* | | |
| Non-visceral metastasis | 19 (24.1) | 7 (22.6) |
| Visceral metastasis | 60 (75.9) | 24 (77.4) |
| Prior (neo)adjuvant therapy, *n* (%) | 48 (60.8) | 16 (51.6) |
| Anthracycline | 38 (48.1) | 13 (41.9) |
| Taxane | 33 (41.8) | 8 (25.8) |
| Trastuzumab | 24 (30.4) | 6 (19.4) |

*ITT* intention-to-treat, *ER* estrogen receptor, *PgR* progesterone receptor, *ECOG PS* Eastern Cooperative Oncology Group performance status.

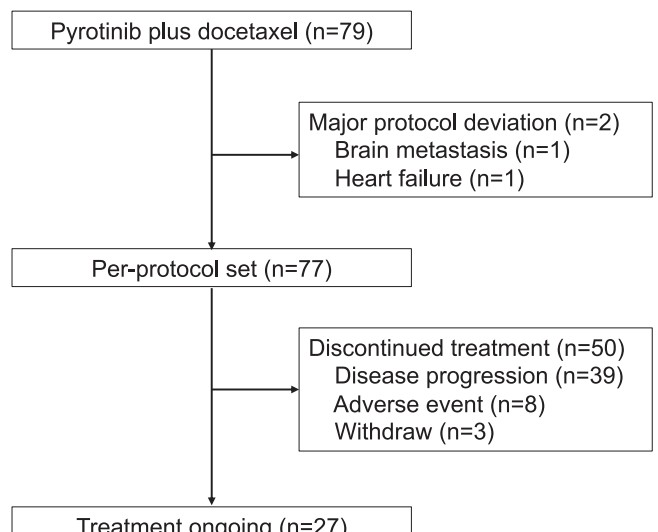

**Fig. 1 | Patient enrollment and follow-up flowchart.** This figure illustrates the flow of patients through the study, including enrollment, allocation, follow-up, and analysis phases. Source data are provided with this paper.

**Table 2 | Tumor response**

| Response | Pyrotinib plus docetaxel | | |
|---|---|---|---|
| | ITT (*n* = 79) | PP (*n* = 77) | Biomarker set (*n* = 31) |
| *Best overall response, n (%)* | | | |
| Complete response | 3 (3.8) | 3 (3.9) | 3 (9.7) |
| Partial response | 60 (75.9) | 60 (77.9) | 25 (80.6) |
| Stable disease | 11 (13.9) | 11 (14.3) | 1 (3.2) |
| Progressive disease | 0 | 0 | 0 |
| Not evaluable | 5 (6.3) | 3 (3.9) | 2 (6.5) |
| Objective response, *n* (%) | 63 (79.7) | 63 (81.8) | 28 (90.3) |
| 95% CI | 70.8–88.6 | 73.2–90.4 | 79.9–100.0 |
| Clinical benefit, *n* (%) | 69 (87.3) | 69 (89.6) | 28 (90.3) |
| 95% CI | 80.0–94.6 | 82.8–96.4 | 79.9–100.0 |
| Time to response (months), median (IQR) | 1.5 (1.4–1.7) | 1.5 (1.4–1.7) | 1.5 (1.38–1.61) |
| Duration of response (months), median (IQR) | 15.9 (8.3–19.5) | 15.9 (8.3–19.5) | 12.4 (8.45–16.53) |

*IQR* interquartile range, *ITT* intention-to-treat, *PP* per-protocol.

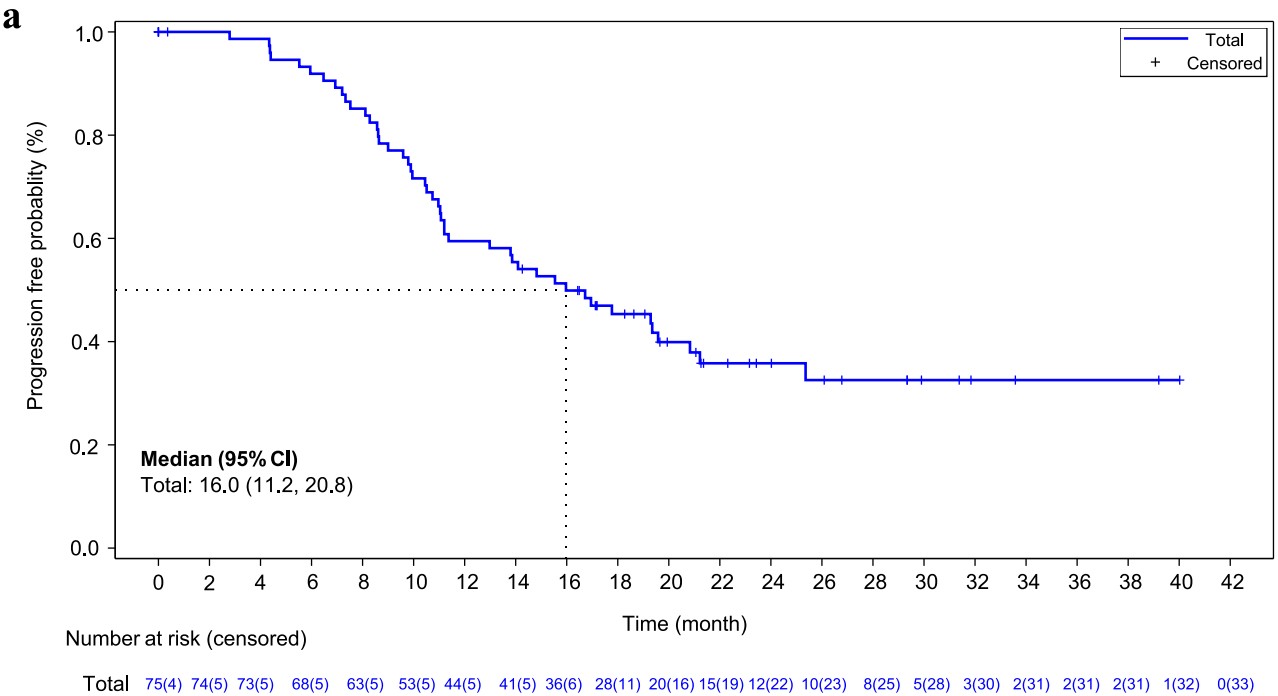

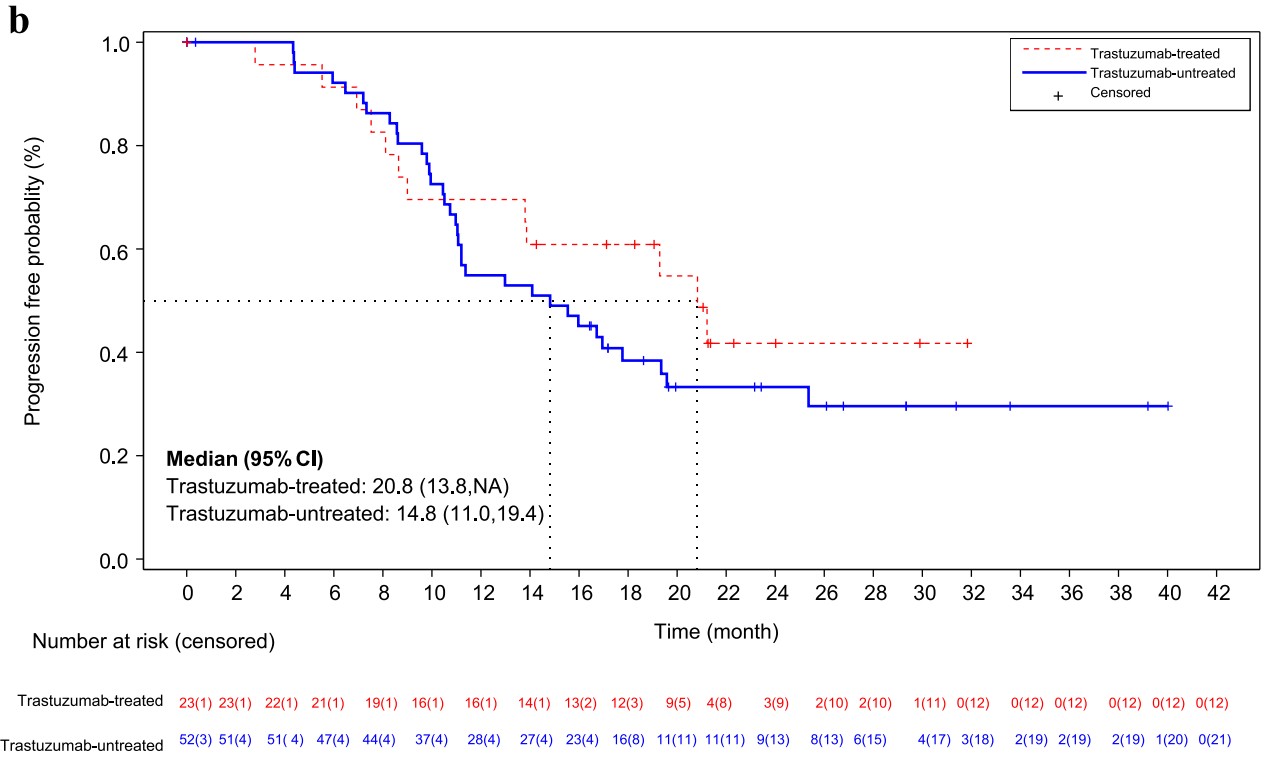

**Fig. 2 | Progression-free survival analysis. a** Progression-free survival among all study participants (n=79). **b** Progression-free survival in trastuzumab-treated and untreated patients. Source data are provided with this paper.

and 89.6% in the ITT and PP populations, respectively (Table 2). The combination of pyrotinib and docetaxel demonstrated rapid and durable responses, with a median time to response (TTR) of 1.5 months and a median duration of response (DoR) of 15.9 months.

As of October 31, 2022, the median follow-up time was 22.3 months, ranging from 19.7 to 26.8 months. The median PFS for all patients was 16.0 months (95% CI, 11.2-20.8). In the subgroup analysis, the median PFS was 20.8 months (95% CI, 13.8-not reached) in 22 trastuzumab-treated patients and 14.8 months (95% CI, 11.0-19.4) in 51 trastuzumab-untreated patients (Fig. 2). Additionally, six patients (7.6%) developed central nervous system (CNS) metastasis as the first site of disease progression. However, data on overall survival (OS) was still immature at the time of analysis.

## Safety

All patients experienced treatment-related adverse events (TRAEs), with 42 patients (53.2%) encountering grade ≥3 TRAEs. Eight patients (10.1%) discontinued treatment, and 20 patients (25.3%) had their doses reduced due to TRAEs. Importantly, no TRAE resulted in any fatalities. The most frequently observed TRAE was diarrhea (62.0%), followed by anemia (56.7%) and leukopenia (51.9%). Among the grade ≥3 TRAEs, the most common were leukopenia (29.1%), neutropenia (27.8%), and diarrhea (21.5%) (Table 3).

It is noteworthy that the occurrence of grade ≥3 diarrhea was notably lower in patients who received loperamide prophylaxis (8.9%; 4/45) compared to those who did not (38.2%; 13/34) (Supplementary Table. 1). Besides, a smaller proportion of patients receiving loperamide prophylaxis experienced treatment discontinuation (6.7%, 3/45) and dose reduction (17.8%, 8/45) in contrast to those who did not

receive the prophylaxis, where the rates were 14.7% (5/34) and 35.3% (12/34), respectively. Additionally, the incidence of grade ≥3 leukopenia and/or grade ≥3 neutropenia was lower in patients who received prophylaxis with pegylated recombinant human granulocyte colony-stimulating factor (PEG-rhG-CSF) (23.3%; 10/43) in comparison to those without prophylaxis (36.1%; 13/36) (Supplementary Table 2).

## Biomarker analysis

The biomarker analysis conducted in this study, including the examination of gene mutations, gene amplification, and microsatellite instability, was predefined in our study protocol. A total of 31 patients were included in the biomarker analysis (Table 1). The mutation landscape of driver genes in these patients was depicted in Supplementary Fig. 1. Among the mutated genes, *TP53* (48.4%) and *PIK3CA* (38.7%) were the most commonly observed. In particular, patients harboring *PIK3CA* mutations showed a numerically shorter PFS compared to patients with *PIK3CA* wild type. Similar trends were observed in other genes related to the *PI3K-AKT-mTOR* pathway, such as *PTEN/AKT*. However, no significant difference in PFS was noted between patients with *TP53* mutations and those with *TP53* wild type. Additionally, patients with *CCNE1* and *IGF1R* amplification tended to have shorter PFS. Interestingly, no association between microsatellite instability score or tumor mutation burden and PFS was observed (Table 4). Furthermore, there was no significant association found between gene mutations, gene amplification, microsatellite instability score, or tumor mutation burden, and ORR (Supplementary Table 3). Moreover, *MYC* amplification was more frequently detected in patients who received prior trastuzumab and in patients with recurrent or metastatic disease (Supplementary Table 4).

## Discussion

Our findings demonstrate that the combination of pyrotinib with docetaxel shows promising antitumor activity and provides a PFS benefit in patients with HER2-positive MBC when used as a first-line treatment. Importantly, no new safety concerns were observed during the study. Notably, the oral formulation of pyrotinib offers a significant advantage over trastuzumab, which requires intravenous injection. The convenience of oral administration is particularly beneficial for patients, especially after completing the docetaxel treatment. In our study, several patients have been on maintenance pyrotinib for over 3 years, suggesting that the ease of oral administration may improve patient adherence to the treatment regimen.

Dual-targeted therapy has become the standard first-line treatment for patients with MBC[1]. The CLEOPATRA study provided evidence supporting the superiority of pertuzumab plus trastuzumab and docetaxel over trastuzumab plus docetaxel in terms of PFS and OS[18]. The Chinese bridging PUFFIN study also confirmed the benefits of dual-targeted therapy in untreated HER2-positive MBC[19,20]. However,

### Table 3 | Treatment-related adverse events occurring in at least 5% of patients

| Event, n (%) | Pyrotinib plus docetaxel (n = 79) | |
|---|---|---|
| | **Any grade** | **≥Grade 3** |
| All | 79 (100%) | 42 (53.2%) |
| Diarrhea | 49 (65.8%) | 17 (21.5%) |
| Anemia | 44 (55.7%) | 3 (3.8%) |
| Leukopenia | 41 (51.9%) | 22 (27.8%) |
| Neutropenia | 33 (41.8%) | 23 (29.1%) |
| ALT/AST increased | 27 (35.4%) | 0 |
| Hypokalemia | 25 (31.6%) | 5 (6.3%) |
| Vomiting | 25 (31.6%) | 1 (1.3%) |
| Hypoalbuminemia | 17 (21.5%) | 0 |
| Hypoproteinemia | 14 (17.7%) | 0 |
| Rash | 14 (17.7%) | 2 (2.5%) |
| Nausea | 13 (16.5%) | 0 |
| Thrombocytopenia | 11 (13.9%) | 1 (1.3%) |
| Stomatitis | 10 (12.7%) | 0 |
| Urinary infection | 10 (12.7%) | 0 |
| Bilirubin increased | 10 (12.7%) | 0 |
| Creatinine increased | 9 (11.4%) | 0 |
| Fatigue | 7 (8.9%) | 1 (1.3%) |
| Hypertriglyceridemia | 7 (8.9%) | 0 |
| Lymphopenia | 6 (7.6%) | 2 (2.5%) |
| Alkaline phosphatase increased | 5 (6.3%) | 0 |

*ALT* alanine transaminase, *AST* aspartate transaminase.

### Table 4 | The associations between biomarkers and progression-free survival

| Gene | No. of patients (WT) | No. of patients (MUT) | mPFS (WT) | mPFS (MUT) | HR (ref = WT) | Exact *P* | Adjusted *P* |
|---|---|---|---|---|---|---|---|
| *PIK3CA* | 17 | 11 | 16.72 (11.2–NR) | 10.4 (9.8–NR) | 2.0547 (0.872–4.8414) | 0.0931 | 0.3865 |
| *TP53* | 15 | 13 | 13.8 (10.4–NR) | 13.0 (8.6–NR) | 1.0374 (0.4462–2.4117) | 0.9321 | >0.9999 |
| Gene | No. of patients (WT) | No. of patients (Amp) | mPFS (WT) | mPFS (Amp) | HR (ref = WT) | Exact *P* | Adjusted *P* |
| *MYC* | 24 | 4 | 14.45 (10–21.2) | 10.8 (8.6–NR) | 2.3625 (0.7466–7.4761) | 0.1317 | 0.4829 |
| *ERBB2* | 2 | 26 | 12.15 (6.5–NR) | 13.4 (10–21.2) | 0.5613 (0.1288–2.4473) | 0.4358 | 0.7414 |
| Feature | No. of patients (Low) | No. of patients (High) | mPFS (Low) | mPFS (High) | HR (ref = Low) | Exact *P* | – |
| MSI score | 9 | 19 | 13 (9.9–NR) | 13.8 (10–NR) | 0.5573 (0.229–1.3562) | 0.1914 | – |
| TMB | 13 | 15 | 13.8 (10.7–NR) | 10.4 (9.8–NR) | 1.1332 (0.4875–2.6343) | 0.7712 | – |

In 31 patients eligible for biomarker analysis, 28 were evaluable. *WT* wild type, *MUT* mutation, *mPFS* median progression-free survival, *HR* hazard ratio, *NR* not reached, *Inf* infinity, *Amp* amplification, *MSI* microsatellite instability, *TMB* tumor mutation burden. Two-sided log-rank test was performed in this analysis, followed by multiple testing correction via the method of Benjamini & Hochberg.

the use of anti-HER2 TKIs in the first-line treatment of HER2-positive MBC has been a subject of ongoing debate. Studies comparing lapatinib plus taxanes to trastuzumab plus taxanes showed that the former resulted in shorter PFS (median PFS: 11.3 months vs. 9.0 months) and raised more safety concerns for untreated HER2-positive MBC patients[4]. Similarly, neratinib plus paclitaxel did not demonstrate superiority over trastuzumab plus paclitaxel in first-line treatment, with both arms showing a median PFS of 12.9 months[5]. Prior research highlighted the superiority of pyrotinib over lapatinib when combined with capecitabine in patients previously treated for HER2-positive MBC[15,16]. However, the efficacy and safety of pyrotinib plus chemotherapy as a first-line treatment for HER2-positive MBC remained unexplored before our study. Our investigation aimed to address this knowledge gap and shed light on the potential role of pyrotinib plus chemotherapy in the first-line treatment of HER2-positive MBC. In our study, the combination of pyrotinib and docetaxel resulted in a confirmed ORR of 79.7% and a median PFS of 16.0 months. Furthermore, the treatment demonstrated rapid and durable responses, with a median TTR of 1.5 months and a median DoR of 15.9 months. These findings provided preliminary efficacy and safety data that paved the way for the initiation of the phase III PHILA study. Notably, the phase III PHILA study confirmed that pyrotinib plus trastuzumab and docetaxel significantly prolonged PFS compared to placebo plus trastuzumab and docetaxel (median PFS: 24.3 months vs. 10.4 months) in the first-line treatment of HER2-positive MBC patients[17]. These results further demonstrate the potential benefits of pyrotinib in the first-line setting for HER2-positive MBC patients.

The therapeutic landscape for HER2-positive MBC is undergoing significant transformation, propelled by the advent of advanced agents such as ADCs, exemplified by trastuzumab deruxtecan[9,10], and TKIs. The DESTINY-Breast09 trial (NCT04784715) is examining trastuzumab deruxtecan, either alone or alongside trastuzumab, against a conventional frontline regimen of chemotherapy, trastuzumab, and pertuzumab for HER2-positive MBC. Concurrently, the HER2CLIMB-05 trial (NCT05132582) is exploring the addition of tucatinib to the standard chemotherapy-trastuzumab-pertuzumab combination. The results of these investigative endeavors, however, remain pending. Moreover, there's burgeoning interest in synergizing ADCs and TKIs for therapeutic amplification. This is underscored by several ongoing studies targeting HER2-positive breast cancer: tucatinib plus T-DM1 in second-line and adjuvant settings (NCT03975647, NCT04457596); tucatinib plus trastuzumab deruxtecan in later-line settings (NCT04539938); SHR-A1811, a HER-2 specific ADC, plus pyrotinib, in a second-line setting (NCT05353361). As we venture into the future, this innovative ADC plus TKI model may soon permeate first-line therapeutic options. The interplay between TKIs and both large molecule monoclonal antibodies and small molecule ADCs signifies an imperative area for continued exploration, where the selection of a robust TKI could be crucial. Within this context, our study has elucidated that pyrotinib might be a viable option for patients with HER2-positive MBC.

Approximately 30-40% of patients with HER2-positive MBC develop CNS metastases after trastuzumab treatment[21,22]. Trastuzumab has been associated with a higher risk of CNS metastases as the first site of disease progression, as indicated by a systematic review[23]. The post-hoc analysis of the CLEOPATRA study demonstrated that the incidence of CNS metastases as the first site of disease progression was 13.7% in the dual-targeted group and 12.6% in the trastuzumab group[24]. However, neratinib has shown promise in reducing the incidence of CNS metastases (8.3% vs. 17.3%) compared to trastuzumab[5]. In our study, six patients (7.6%) developed CNS metastasis as the first site of disease progression. These findings are noteworthy, as they suggest that pyrotinib, as demonstrated in the PERMEATE study, may be effective in delaying CNS metastasis and treating intracranial diseases in patients with HER2-positive MBC[25]. Such results indicate the

potential of pyrotinib as a valuable option to manage CNS metastases, which is a crucial aspect of treatment in this patient population.

In our study, more than 30% of patients had received trastuzumab in their (neo)adjuvant setting, which was a higher proportion compared to previous studies[4–6,19]. Notably, in the CLEOPATRA study and PERSUE study, trastuzumab-treated patients showed a shorter PFS than trastuzumab-untreated patients[6,26]. However, the results from both our study and the PHILA study demonstrated that pyrotinib led to a longer PFS in trastuzumab-treated patients compared to trastuzumab-untreated patients[17]. In our study, trastuzumab-treated patients had a median PFS of 20.8 months, while trastuzumab-untreated patients had a median PFS of 14.8 months. This observation suggests a potential synergistic effect between HER2 TKIs and monoclonal antibodies. Preclinical studies have indicated that the combination of HER2 TKIs and trastuzumab can effectively suppress the growth of breast cancer cells[27]. The use of HER2 TKIs may also overcome potential resistance to trastuzumab, leading to improved treatment outcomes[28].

Diarrhea has been a common AE associated with the use of small molecule anti-HER2 TKIs such as lapatinib, pyrotinib, and neratinib[4,5,14,15]. The development of diarrhea can significantly impact the quality of life for patients and may even lead to dose reductions or treatment interruptions, which can compromise the effectiveness of the treatment[29]. The CONTROL study demonstrated that proactive management strategies, such as primary prophylaxis or dose escalation, can enhance the tolerability of neratinib, reduce the incidence and severity of diarrhea, and decrease treatment discontinuations due to diarrhea[30]. In our study, we implemented loperamide as primary prophylaxis to manage potential diarrhea, resulting in significantly lower incidence rates of grade ≥3 diarrhea in patients who received prophylaxis (8.9%; 4/45) compared to those who did not (38.2%; 13/34). Furthermore, patients who received loperamide prophylaxis had a lower incidence of treatment discontinuation (6.7%, 3/45) and dose reduction (17.8%, 8/45) compared to those without prophylaxis, who had rates of 14.7% (5/34) and 35.3% (12/34) respectively. Chemotherapy-induced neutropenia is another serious concern in the treatment of patients with MBC. To address this issue, short- and long-acting granulocyte colony-stimulating factors (G-CSFs) are commonly used. According to a phase III trial, PEG-rhG-CSF has shown to be more effective and safer in preventing neutropenia than daily rhG-CSF in breast cancer patients receiving chemotherapy[31]. In our study, 43 patients received PEG-rhG-CSF prophylaxis, and these patients experienced a lower likelihood of developing grade ≥3 leukopenia and/or neutropenia compared to those who did not receive this prophylaxis.

Previous studies have consistently demonstrated that patients harboring PIK3CA mutations exhibit a worse prognosis compared to those with PIK3CA wild type when treated with anti-HER2 agents. The CLEOPATRA study, which included both the pertuzumab group and the control group, found that PIK3CA mutation was associated with a shorter PFS[32]. Preclinical research has also indicated that mutated PIK3CA can lead to resistance to lapatinib[33]. Similarly, in the neoadjuvant setting, PIK3CA mutations were associated with a lower pathological complete response rate in patients receiving neoadjuvant pyrotinib, trastuzumab plus chemotherapy[34]. Our study's findings are consistent with these observations, as patients with PIK3CA mutations in our study also showed a numerically shorter PFS compared to those with PIK3CA wild type. Additionally, similar trends were observed in other genes related to the PI3K-AKT-mTOR pathway, including PTEN/AKT. A previous phase I study of pyrotinib for patients with HER2-positive MBC also reported significantly longer PFS in patients with wild type PIK3CA and TP53 compared to those with PIK3CA or TP53 mutations[35]. However, in our first-line setting, the PFS was comparable between patients with TP53 mutation and those with wild type. The role of TP53 mutation in the prognosis of MBC warrants further

evaluation. Given the small sample size in our study, these results need to be interpreted with caution. However, investigating the resistance pathways to trastuzumab and identifying potential drugs that can reverse trastuzumab resistance remain crucial areas for future research.

This study has several limitations that should be considered when interpreting the results. Firstly, it is a single-arm study with a limited sample size, which may introduce potential bias and limit the generalizability of the findings. Secondly, the study only enrolled Chinese patients, and the efficacy and safety of pyrotinib plus docetaxel in other populations remain unknown. Further research in diverse populations is needed to validate the results. Thirdly, the OS data was still immature at the time of analysis, and longer-term follow-up is ongoing. Fourthly, the statistical significance between biomarkers and treatment outcomes was challenging to detect due to the small number of patients suitable for biomarker analysis. Additionally, some mutations were infrequently detected, and caution should be exercised when interpreting these results.

Pyrotinib in combination with docetaxel demonstrated promising antitumor activity and showed a significant PFS benefit for patients with HER2-positive MBC in the first-line setting. The use of loperamide as prophylaxis effectively reduced the incidence of diarrhea, while prophylactic PEG-rhG-CSF administration was successful in preventing leukopenia and/or neutropenia. Future research with larger and more diverse patient populations will be essential to validate and expand on these promising results.

## Methods

### Study design and patients
The study was conducted in accordance with the Declaration of Helsinki and Good Clinical Practice principles. The protocol was approved by the ethics committee of Zhejiang Cancer Hospital, Sun Yat-sen University Cancer Center, Harbin Medical University Cancer Hospital, Nanchang People's Hospital, Fujian Cancer Hospital, The First Affiliated Hospital, Zhejiang University School of Medicine, The Second Affiliated Hospital, Zhejiang University School of Medicine, Sir Run Run Shaw Hospital, Zhejiang University School of Medicine, Beijing Cancer Hospital and The Second Affiliated Hospital of Dalian Medical University. All patients provided written informed consent before any procedure. The study was preregistered with clinical trial.gov on March 2019 with the registration number NCT03876587. Patients were provided with the study treatment for free and also received a transportation reimbursement.

PANDORA is a multicenter, single-arm phase II trial with a Simon two-stage design, involving woman with HER2-positive MBC who had not received HER2 blockade or chemotherapy for metastatic disease between June 12, 2019 and June 18, 2021. The trial was conducted at 10 centers in China (Zhejiang Cancer Hospital, Sun Yat-sen University Cancer Center, Harbin Medical University Cancer Hospital, Nanchang People's Hospital, Fujian Cancer Hospital, The First Affiliated Hospital, Zhejiang University School of Medicine, The Second Affiliated Hospital, Zhejiang University School of Medicine, Sir Run Run Shaw Hospital, Zhejiang University School of Medicine, Beijing Cancer Hospital and The Second Affiliated Hospital of Dalian Medical University). Eligible patients met the following criteria: (1) histologically confirmed MBC and candidates for chemotherapy; (2) HER2-positive, defined as immunohistochemistry (IHC) staining 3+ or IHC staining 2+ with fluorescence in situ hybridization positive; (3) aged between 18 and 70 years old; (4) an Eastern Cooperative Oncology Group performance status (ECOG PS) of 0-1; (5) at least one measurable disease according to Response Evaluation Criteria in Solid Tumors (RECIST) version 1.1; (6) may have received hormonal regimens in the metastatic setting; (7) (neo)adjuvant trastuzumab or taxane treatment was permitted, with a disease-free interval of more than 12 months from completion of the taxane and more than 6 months from completion of the trastuzumab.

Patients who had previously received anti-HER2 TKI were excluded, as were those with CNS metastasis. Detailed inclusion and exclusion criteria can be found in the protocol.

### Procedures
Eligible patients received daily oral pyrotinib 400 mg and intravenous docetaxel 75 mg/m$^2$ every 3 weeks until disease progression, unacceptable toxicity, or withdrawal of consent. Docetaxel treatment was administered for at least six cycles, and patients were allowed to discontinue docetaxel at their discretion or as decided by the investigators after six cycles. Dose reductions of docetaxel (first reduction: 60 mg/m$^2$; second reduction: 50 mg/m$^2$) and pyrotinib (first reduction: 320 mg/d; second reduction: 240 mg/d) were permitted based on safety profiles.

Due to the high incidence of diarrhea, leukopenia, and/or neutropenia observed during the first stage of the study, the study protocol was modified during the second stage. Primary prophylaxis for diarrhea with loperamide was recommended for the first six weeks of pyrotinib treatment (day 1 to day 14: 2 mg three times daily; day 14 to day 42: 2 mg twice daily). Additionally, subcutaneous PEG-rhG-CSF at a dose of 6 mg or 100 µg/kg was recommended 24-48 h before docetaxel dosing to prevent leukopenia and/or neutropenia.

Tumor assessments using enhanced computed tomography or magnetic resonance imaging were performed every 6 weeks until disease progression or death, following the RECIST version 1.1. The response had to be confirmed in the next assessment. Any AEs during the study were recorded and graded according to the National Cancer Institute Common Terminology Criteria for Adverse Events (NCI-CTCAE) version 5.0.

### Endpoints
The primary endpoint of this study was the ORR, defined as the proportion of patients who achieved a confirmed CR or PR. Secondary endpoints included OS, which was defined as the time from obtaining informed consent to death from any cause, and PFS, defined as the time from obtaining informed consent to disease progression or death from any cause, whichever came first. Other secondary endpoints were the CBR, defined as the proportion of patients with confirmed CR, PR, or stable disease (SD) lasting for at least 24 weeks, and the DoR, defined as the time from the first documented response to disease progression for patients with confirmed CR and PR. Safety profiles were also assessed as a secondary endpoint. Additionally, exploratory endpoints included the TTR, which was calculated as the time from treatment initiation to the first documented response, and the incidence of CNS metastasis as the first site of disease progression.

### Samples and panel design
Biomarker analysis was conducted using a capture-based sequencing panel targeting 561 genes. DNA was extracted form formalin fixed paraffin embedded tumor sample. The capture-based sequencing panel used in this study was designed and provided by Precision Scientific (Beijing) Co., Ltd. This panel comprises 561 genes frequently mutated in solid tumors and closely associated with precision therapy in cancers, including 201 cancer driver genes, 51 cancer predisposition genes, 44 genes appearing in key DNA damage repair pathways, 174 genes possessing actionable mutation spots of targeted therapy and 28 immuno-oncology genes. Probes of this panel cover exonic regions of all genes and hotspots in intronic or promoter regions of part of them, enabling the comprehensive analysis for mutational profiling of cancers.

### Somatic mutation detection
Sequencing reads were first processed using fastp (v0.18.1) for adapter trimming and quality filtering[36]. Clean reads were aligned to the human reference genome (hg19) using bwa 0.7.17[37], and aligned reads were

then sorted using sambamba (v0.6.6)[38]. Duplicates were marked via Picard (v1.122, https://broadinstitute.github.io/picard/). To improve the alignment accuracy, the Genome Analysis Toolkit (GATK, v 4.2.0.0) was used to process BAM files resulted from the previous step through steps including local realignment around indels and base quality recalibration[39]. Next, Mutect2 was used for base substitution calling with the panel of normals (PON) provided in GATK resource bundle and IndelGenotyper was used for InDels calling both in single-sample mode. We only retained mutations covered by more than 10 reads in total and more than 2 reads supporting the alternative allele. The ratio of reads supporting the alternative allele should be more than 0.02. Following that, retained mutations were annotated using ANNOVAR[40]. Mutations with population prevalence larger than 0.01 in either of ExAC and 1000 Genomes database were further filtered out. Dysfunctional mutations in exonic region and at splicing junction sites were retained for downstream analysis. Mutation in driver genes of breast cancer[41] was highlighted in the final mutational landscape.

### Copy number variation (CNV) detection
CNVkit (v0.8.0) was adopted to build a reference of copy number profile from 328 blood samples collected from breast cancer patients as the control[42]. CNV value of genes was then calculated for each sample involved in this study based on the constructed reference and BAM file recording the aligned sequencing reads using CNVkit. Amplifications and deletions were determined if the CNV value is higher than 0.3 and lower than −0.4, respectively. Only amplifications and deletions happened in driver genes were retained for downstream analysis[43].

### Tumor mutation burden (TMB) and microsatellite instability (MSI) score
First, dysfunctional mutations in exonic region and at splicing junction sites with variant frequency not smaller than 0.05 were counted. Then, the TMB value was calculated as the panel size normalized number of counted mutations. TMB was determined by dividing the total number of identified mutations by the panel size (2MB). MSI score was calculated using MSIsensor-pro (v1.0.2) based on BAM file recording the aligned sequencing reads[44].

### Statistics analysis
The study was conducted using a two-stage Simon design. Based on previous data of trastuzumab and docetaxel as a first-line treatment in HER2-positive MBC, the null hypothesis for the ORR was set at 60%[3], while the alternative hypothesis was an ORR of 75%. With a one-sided α of 0.05 and a power of 0.80, the first stage required 27 evaluable patients. If more than 17 patients achieved responses in the first stage, an additional 40 evaluable patients would be enrolled in the second stage. If more than 46 out of 67 patients achieved responses in total, the results were considered positive. Considering a dropout rate of 15%, a total of 79 patients were required for this study. The sample size was calculated using NCSS&PASS version 15.0.

The ITT population included all patients who received at least one dose of pyrotinib and docetaxel. The PP population included patients who received at least one dose of pyrotinib and docetaxel and did not have any major protocol violations. Efficacy was analyzed in both ITT and PP populations. The safety analysis set included patients who received at least one dose of pyrotinib and docetaxel and had safety records. The biomarker set included patients who received at least one dose of pyrotinib and were evaluable for biomarker analysis.

Continuous data were presented as median and range or interquartile range (IQR), while categorical data were described as numbers and percentages. The 95% CI of ORR and CBR were estimated using the Clopper-Pearson method. For time-to-event data, the Kaplan–Meier method was utilized, and the median time and their 95% CIs were estimated. These analyses were conducted using SAS version 9.3. For the biomarker analysis, Wilcoxon rank sum test was applied to compare the difference of continuous variable between sample groups, while Fisher's exact test was applied to test for the independence between categorical variables. Univariate Cox regression model and log-rank test wrapped in the R package named survival (v3.3.1) were used to perform survival analysis. Multiple testing correction was performed using the method of Benjamini & Hochberg. A two-sided $P < 0.05$ was deemed statistically significant. Visualization of biomarkers was performed via the R package named ggplot2 (v3.3.6).

### Reporting summary
Further information on research design is available in the Nature Portfolio Reporting Summary linked to this article.

## Data availability
The trial protocol is provided in the Supplementary Information file. Deidentified clinical data for individual patients supporting the results of this manuscript are included within the manuscript and its Supplementary files. To ensure patient privacy, further deidentified data are not publicly available, but can be accessed upon a scientifically justified request to the corresponding author, Xiaojia Wang (wxiaojia0803@163.com), for up to 10 years after this paper's publication. The corresponding author will review all data requests to ensure its scientific use, and upon approval, de-identified patient data will be shared within three months. The raw sequence data reported in this paper have been deposited in the Genome Sequence Archive in National Genomics Data Center, China National Center for Bioinformation/Beijing Institute of Genomics, Chinese Academy of Sciences (GSA-Human: HRA005624) and are publicly accessible at https://ngdc.cncb.ac.cn/gsa-human/browse/HRA005624. The remaining data are available within the Article, Supplementary Information or Source Data file. Source data are provided with this paper.

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

## Acknowledgements

This study was supported by Jiangsu Hengrui Pharmaceuticals Co., Ltd. The funding organizations had no role in the design and conduct of the study; collection, management, and analysis of the data; review or approval of the manuscript; and decision to submit the manuscript for publication. We thank Silu Wang (Department of Medical affairs, Jiangsu Hengrui Pharmaceuticals Co., Ltd.) for his assisstence in data interpretation, Yitao Wang (Department of Medical affairs, Jiangsu Hengrui

Pharmaceuticals Co., Ltd.) for his assistance in statistical support, and Zhongjiang Chen (Department of Medical affairs, Jiangsu Hengrui Pharmaceuticals Co., Ltd.) for medical writing assistance.

## Author contributions

X.W. and Z.C. had full access to all the data in the study and takes responsibility for the integrity of the data and the accuracy of the data analysis. Y.Z. and W.-M.C. contributed equally to this article. Concept and design: X.W., Z.C. Acquisition, analysis, or interpretation of data: Y.Z., W.-M.C., X.S., Y.S. l.C., W.C., J.L., P.S., Y.C., X.W., H.L., M.L., Z.C., X.W. Drafting of the paper: Y.Z., W.-M.C. Critical revision of the paper for important intellectual content: All authors. Statistical analysis: Y.Z., W.-M.C. Obtained funding: X.W. Administrative, technical, or material support: X.W., Z.C. Supervision: X.W., Z.C.

## Competing interests

The authors declare no competing interests.
