## [Peer Review File · Nature Communications]

Pyrotinib plus docetaxel as first-line treatment for HER2-positive metastatic breast cancer: the PANDORA phase II trialREVIEWER COMMENTS

Reviewer #1 (Remarks to the Author): with expertise in biostatistics, clinical trial study design

The statistical plan, including sample size justification and data analysis, was consistent between the manuscript and the protocol. A Simon two-stage design was employed, determining a sample size of 27 patients in the 1st stage and an additional 40 evaluable patients in the 2nd stage, resulting in a total of 79 patients after accounting for a 15% dropout rate.

Various statistical concerns need clarification.

1. Should the study stop enrollment after 67 evaluable patients based on the planned Simon two-stage design? The study enrolled a total of 79 patients. Did the study deviate the protocol, raising ethical concern?
2. The study modified the trial in the 2nd stage due to the high toxicity rate. Did the toxicity rate be mitigated in the 2nd stage? Please provide comparison of toxicity rate between the 1st and 2nd stage for TRAE and grade ≥ 3 .
3. Please clarify and justify if duration of response includes the time from treatment initiation to disease progression for patients with SD. “(DoR, defined as the time from the first documented response to disease progression for patients with confirmed CR and PR, and the time from treatment initiation to disease progression for patients with SD)”.
4. Various issues in biomarker analysis.
 - a. Are baseline characteristics of patients comparable between the biomarker analysis cohort (n=31) and the entire cohort (n=79)? If not, would the result be biased? What is the response distribution in the biomarker analysis cohort?
 - b. Discrepancy between Supplementary Figure 1 (Mutation landscape of driver genes) and Supplementary Table 2 (The associations between biomarkers and objective response rate). For example, there were BCL2 mutations in 2 patients with PR in Supplementary Figure 1. In contrast, Supplementary Table 2 had BCL2 mutation only in one patient with CR.
 - c. HR in Table 4 is questionable with n=1-2 in mutation group. Did the mutations occur in same patient?
 - d. OR in Supplementary Table 2 and 3 is also questionable due to small sample size.

Reviewer #2 (Remarks to the Author): with expertise in breast cancer, therapy

In this article, the authors report the results from a multicentric single arm phase II trial in patients treated for metastatic HER2-amplified breast cancer and naïve from HER2 targeted therapies and chemotherapy. In this clinical trial, patients were treated in 1st line with chemotherapy (docetaxel 75 mg/m² every 3 weeks) + daily oral pyrotinib (a new oral tyrosine kinase inhibitor of HER1, 2 and 4).

The primary endpoint was overall response rate (ORR)

- 79 patients were enrolled in this study
- ORR was 79.7%
- Median PFS was 16 months with an acceptable safety profile (extensively described in the manuscript)
- No OS data are reported yet

The results presented in this paper are about baseline clinical characteristics of the 79 patients included, data concerning best tumor response obtained according to radiological RECIST criteria, PFS data and adverse events.

The only translational study reported in the manuscript is the result from the sequencing panel targeting 561 genes frequently mutated in human cancers (driver genes) and showing association with mutation and PFS.

The concept of this study is clinically interesting and proves the possibility of combining IV taxane chemotherapy with an oral TKI targeting HER2.

however, this study does not bring anything new to the current landscape of treatment of HER2-amplified breast cancer:

The current standard of care in first-line treatment is chemotherapy (taxane) + trastuzumab and pertuzumab (CLEOPATRA trial: mPFS 19 months, with a major benefit in terms of OS), and the results of this phase 2 will not change this standard of care.

In addition, the recent arrival of new molecules showing extremely promising results, with meaningful benefits in terms of overall survival, such as ADCs or other TKIs such as tucatinib, are undergoing significant development with current first-line trials.

Finally, the results reported here concern a small group of patients, included in a phase II trial with a new molecule, whose development is limited to a certain part of the world only,

and which, despite encouraging results, does not seem to bring a significant benefit compared to current standards, and to new strategies in development such as ADCs targeting HER2.

On the other hand, the lack of new translational biological results makes me think that this article should be submitted to a clinical oncology journal and not to a journal such as Nature Communication which does not seem to be adapted to the scientific, biological and innovative content of this publication.

Reviewer #3 (Remarks to the Author): with expertise in breast cancer, therapy

This is a phase study aimed to investigate the efficacy and safety of pyrotinib plus docetaxel as first-line treatment in HER2-positive MBC. The confirmed ORR was 79.7% (95% confidence interval [CI], 70.8-88.6) in the intention-to-treat population. The median PFS was 16.0 months (95% CI, 11.2-20.8). This trial is of interest, since more than 30% of patients had received trastuzumab in their (neo)adjuvant setting in this study. However, there are several aspects that require further explanation:

- 1) This is a single-arm study with a limited sample size, authors should avoid comparing the efficacy results observed here with the CLEOPATRA study and other phase 3 randomized controlled trials.
- 2) Nearly 70% of patients in the study had not received trastuzumab in their (neo)adjuvant setting, I am concerning about using a small molecule TKI plus docetaxel other than the standard THP for these patients. Moreover, I think it will be more interesting to explore the efficacy and safety of pyrotinib in combination with Herceptin and docetaxel.
- 3) Prior studies using the small molecule TKI pyrotinib showed high percentages of diarrhea, and I have noticed that this study mentioned a modification of study protocol due to a high incidence of diarrhea, leukopenia, and/or neutropenia during the first stage of the study. Thus, this AE also limited the use of pyrotinib in the first-line treatment.
- 4) The introduction section lacks important information to contextualize the current trial.
- 5) 77.2% of the patients enrolled has a ECOG of 1. This is unexpected in a trial enrolling untreated HER2-positive metastatic patients. Authors should explain.
- 6) Authors should improve the language in this study.

Reviewer #1

The statistical plan, including sample size justification and data analysis, was consistent between the manuscript and the protocol. A Simon two-stage design was employed, determining a sample size of 27 patients in the 1st stage and an additional 40 evaluable patients in the 2nd stage, resulting in a total of 79 patients after accounting for a 15% dropout rate.

Various statistical concerns need clarification.

1. Should the study stop enrollment after 67 evaluable patients based on the planned Simon two-stage design? The study enrolled a total of 79 patients. Did the study deviate the protocol, raising ethical concern?

Response: We appreciate your careful review and insightful comment. Our study was indeed initially designed with a Simon two-stage model in mind, and our calculation for a total of 79 patients was based on an expected 15% dropout rate, mainly due to potential AEs such as diarrhea caused by the pyrotinib-docetaxel combination therapy. We designed our study this way to ensure that we would have a sufficient number of evaluable patients by the end of the study. However, during the course of the study, we found that the primary prophylaxis for diarrhea with loperamide was very effective, resulting in a lower dropout rate (6.3%, two due to major protocol violations, and three withdrew from the study due to AEs) than initially expected. Even though the observed dropout rate was lower, we decided to keep our predefined sample size unchanged. The rationale for this decision was two-fold: First, adjusting the sample size during the study based on interim findings could potentially introduce bias into the study. Second, the protocol, including the sample size, was thoroughly reviewed and approved by the ethical review board of all participating centers. In conclusion, we believe our adherence to the originally approved protocol did not raise any ethical concerns, as our decision to continue enrollment up to our predefined sample size was based on the principle of maintaining the validity and integrity of our study.

2. The study modified the trial in the 2nd stage due to the high toxicity rate. Did the toxicity rate be mitigated in the 2nd stage? Please provide comparison of toxicity rate between the 1st and 2nd stage for TRAE and grade ≥ 3 .

Response: Thank you for your thoughtful question. We acknowledge that the trial did introduce primary prophylaxis measures with loperamide and PEG-rhG-CSF in the second stage to

manage the initially high toxicity rates. However, not all patients in the second stage received this prophylaxis. Given the variation in prophylaxis uptake, we believe it would be more informative to compare the incidence of toxicity in patients who received the prophylaxis versus those who did not. We have included this information in our supplementary materials now (new Supplementary Table 1 and Table 2). To elaborate: The occurrence of grade ≥ 3 diarrhea was less frequent in patients who received loperamide prophylaxis (8.9%; 4/45) compared to those who did not (38.2%; 13/34). Additionally, a smaller proportion of patients receiving loperamide prophylaxis experienced treatment discontinuation (6.7%, 3/45) and dose reduction (17.8%, 8/45) in contrast to those who did not receive the prophylaxis, where the rates were 14.7% (5/34) and 35.3% (12/34), respectively. The incidence of grade ≥ 3 leukopenia and/or grade ≥ 3 neutropenia was less common in patients who received PEG-rhG-CSF prophylaxis (23.3%; 10/43) than in those who did not (36.1%; 13/36). From these observations, we can infer that the primary prophylaxis measures implemented in the second stage were successful in mitigating the toxicity rate of the treatment.

3. Please clarify and justify if duration of response includes the time from treatment initiation to disease progression for patients with SD. “(DoR, defined as the time from the first documented response to disease progression for patients with confirmed CR and PR, and the time from treatment initiation to disease progression for patients with SD)”.

Response: Thank you for your question. Indeed, in our study design, DoR does include the time from treatment initiation to disease progression for patients with SD, which was consistent with our previous studies [1]. It was clarified: the duration of response (DoR), defined as the time from the first documented response to disease progression for patients with confirmed CR and PR, and the time from treatment initiation to disease progression for patients with SD.

4. Various issues in biomarker analysis.

a. Are baseline characteristics of patients comparable between the biomarker analysis cohort (n=31) and the entire cohort (n=79)? If not, would the result be biased? What is the response distribution in the biomarker analysis cohort?

Response: Thank you for your insightful question regarding the comparability of the biomarker analysis cohort and the entire cohort. In response to your query, we have supplemented

our manuscript with the baseline characteristics and tumor response of the 31 patients who underwent biomarker analysis. We have found that both the baseline characteristics and tumor responses are generally comparable between these 31 patients and the entire cohort. For more specific details, please refer to the new Table 1 and Table 2.

b. Discrepancy between Supplementary Figure 1 (Mutation landscape of driver genes) and Supplementary Table 2 (The associations between biomarkers and objective response rate). For example, there were BCL2 mutations in 2 patients with PR in Supplementary Figure 1. In contrast, Supplementary Table 2 had BCL2 mutation only in one patient with CR.

Response: Thank you for your careful observation and thoughtful question. Actually, there is no discrepancy between Supplementary Table 2 (new Supplementary Table 3) and Supplementary Figure 1. The patient with a CR in Supplementary Table 2 indeed had a BCL2 mutation, whereas the two patients with PR in Supplementary Figure 1 had BCL2 amplification, not mutation. Since there were numerous mutated genes, it was not feasible to display all of them in Supplementary Figure 1, hence we opted to show the more prevalent mutated genes. The BCL2 mutation, being present in only one patient, was not included in the figure, which may have led to the misunderstanding. To resolve this confusion and present a more comprehensive picture, we have now updated Supplementary Figure 1 to include the information of BCL2 mutation. We apologize for any confusion caused by the initial presentation, and appreciate your diligence in examining our work.

c. HR in Table 4 is questionable with n=1-2 in mutation group. Did the mutations occur in same patient?

Response: We greatly value your insightful comment. Indeed, these multiple mutations were detected in the same individual, and the primacy of a particular gene mutation has not yet been determined. We fully concur with your concern regarding the limited interpretability of these results, given the small sample size of the mutation group. In line with this, we have endeavored to avoid overinterpreting these data in the Results and Discussion sections of our manuscript. We initially grappled with the decision of whether to present these results and ultimately chose to include them.

In the Discussion, we have acknowledged a limitation: Fourthly, the statistical significance between biomarkers and treatment outcomes was challenging to detect due to the small number of patients suitable for biomarker analysis. Additionally, some mutations were infrequently detected, and caution should be exercised when interpreting these results.

d. OR in Supplementary Table 2 and 3 is also questionable due to small sample size.

Response: Thank you for your comment. We acknowledge your concern about the small sample size, which indeed poses a challenge in generating robust conclusions from the data. Similarly, these results stem from a small subset of patients exhibiting specific gene mutations. Consequently, the limited sample size undoubtedly impacts the strength of the interpretations we can make based on these data. As we previously mentioned, we have been cautious not to overinterpret these results in our manuscript. We appreciate your prudent perspective and understand if you recommend the removal of these data due to the potential for misleading interpretation. In our manuscript's Discussion section, we have underscored this limitation. We are grateful for your continued guidance and welcome further feedback on improving our manuscript.

Reviewer #2

In this article, the authors report the results from a multicentric single arm phase II trial in patients treated for metastatic HER2-amplified breast cancer and naïve from HER2 targeted therapies and chemotherapy. In this clinical trial, patients were treated in 1st line with chemotherapy (docetaxel 75 mg/m² every 3 weeks) + daily oral pyrotinib (a new oral tyrosine kinase inhibitor of HER1, 2 and 4).

The primary endpoint was overall response rate (ORR)

- 79 patients were enrolled in this study

- ORR was 79.7%

- Median PFS was 16 months with an acceptable safety profile (extensively described in the manuscript)

- No OS data are reported yet

The results presented in this paper are about baseline clinical characteristics of the 79 patients included, data concerning best tumor response obtained according to radiological RECIST criteria, PFS data and adverse events.

The only translational study reported in the manuscript is the result from the sequencing panel targeting 561 genes frequently mutated in human cancers (driver genes) and showing association with mutation and PFS.

The concept of this study is clinically interesting and proves the possibility of combining IV taxane chemotherapy with an oral TKI targeting HER2.

however, this study does not bring anything new to the current landscape of treatment of HER2-amplified breast cancer:

The current standard of care in first-line treatment is chemotherapy (taxane) + trastuzumab and pertuzumab (CLEOPATRA trial: mPFS 19 months, with a major benefit in terms of OS), and the results of this phase 2 will not change this standard of care.

Response: We thank you for your valuable input and appreciate the opportunity to clarify the contribution of our study within the existing landscape of HER2-amplified breast cancer treatment. Indeed, we acknowledge that the current standard first-line treatment for HER2-positive MBC is chemotherapy combined with trastuzumab and pertuzumab, as outlined in the CLEOPATRA trial [2-4]. However, the role of anti-HER2 TKIs as a part of the first-line treatment regimen for HER2-

positive MBC has remained a point of contention.

Comparatively, treatment involving lapatinib (another TKI) combined with taxanes showed less promising PFS (median PFS: 9.0 months) and greater safety concerns than the trastuzumab plus taxanes regimen for treatment-naive patients with HER2-positive MBC [5]. Previous studies have shown the superiority of pyrotinib over lapatinib when combined with capecitabine for patients previously treated for HER2-positive MBC [1, 6]. However, the efficacy and safety of pyrotinib plus chemotherapy in the first-line treatment of HER2-positive MBC had not been evaluated before our investigation. Our study helps fill the knowledge gap concerning the role of pyrotinib plus chemotherapy as a first-line treatment for HER2-positive MBC. Indeed, as you correctly pointed out, our phase II study will not change the established standard of care immediately. However, we believe that our findings provide preliminary efficacy and safety data that laid the foundation for the phase III PHILA study, which launched after the commencement of our study. This subsequent study has shown that pyrotinib combined with trastuzumab and docetaxel significantly prolonged PFS compared to placebo plus trastuzumab and docetaxel in the first-line treatment of HER2-positive MBC patients [7].

In summary, our study not only provides new data on the potential of pyrotinib plus docetaxel as a first-line treatment regimen but also offers valuable initial data that helped set the stage for the phase III PHILA study. We hope that our findings and ongoing research will contribute to the development of new, effective treatment strategies for patients with HER2-positive MBC. We appreciate your careful review and valuable feedback, which have helped us clarify the unique contributions of our study.

Following is revised in our Discussion:

Dual-targeted therapy has become the standard first-line treatment for patients with MBC [8]. The CLEOPATRA study provided evidence supporting the superiority of pertuzumab plus trastuzumab and docetaxel over trastuzumab plus docetaxel in terms of PFS and OS [9]. The Chinese bridging PUFFIN study also confirmed the benefits of dual-targeted therapy in untreated HER2-positive MBC [10, 11]. However, the use of anti-HER2 TKIs in the first-line treatment of HER2-positive MBC has been a subject of ongoing debate. Studies comparing lapatinib plus taxanes to trastuzumab plus taxanes showed that the former resulted in shorter PFS (median PFS: 11.3 months vs. 9.0

months) and raised more safety concerns for untreated HER2-positive MBC patients [5]. Similarly, neratinib plus paclitaxel did not demonstrate superiority over trastuzumab plus paclitaxel in first-line treatment, with both arms showing a median PFS of 12.9 months [12]. Prior research highlighted the superiority of pyrotinib over lapatinib when combined with capecitabine in patients previously treated for HER2-positive MBC [1, 6]. However, the efficacy and safety of pyrotinib plus chemotherapy as a first-line treatment for HER2-positive MBC remained unexplored before our study. Our investigation aimed to address this knowledge gap and shed light on the potential role of pyrotinib plus chemotherapy in the first-line treatment of HER2-positive MBC. In our study, the combination of pyrotinib and docetaxel resulted in a confirmed ORR of 79.7% and a median PFS of 16.0 months. Furthermore, the treatment demonstrated rapid and durable responses, with a median TTR of 1.5 months and a median DOR of 15.9 months. These findings provided preliminary efficacy and safety data that paved the way for the initiation of the phase III PHILA study. Notably, the phase III PHILA study confirmed that pyrotinib plus trastuzumab and docetaxel significantly prolonged PFS compared to placebo plus trastuzumab and docetaxel (median PFS: 24.3 months vs. 10.4 months) in the first-line treatment of HER2-positive MBC patients [7]. These results further demonstrate the potential benefits of pyrotinib in the first-line setting for HER2-positive MBC patients.

In addition, the recent arrival of new molecules showing extremely promising results, with meaningful benefits in terms of overall survival, such as ADCs or other TKIs such as tucatinib, are undergoing significant development with current first-line trials.

Response: We appreciate your comments and recognize the exciting progress being made in this area with the development of new therapeutic agents. The landscape of HER2-positive MBC treatment is indeed evolving, with the emergence of ADCs such as trastuzumab deruxtecan and other TKIs such as tucatinib. Trastuzumab deruxtecan have shown great promise due to their ability to deliver cytotoxic drugs directly to HER2-positive cancer cells, thereby maximizing efficacy and minimizing systemic toxicity [13, 14]. Similarly, tucatinib has shown outstanding results, particularly in patients with brain metastases, an area where many HER2-targeted therapies struggle [15, 16]. These agents are currently undergoing significant development, with ongoing trials evaluating their efficacy and safety in first-line settings for HER2-positive metastatic breast cancer.

For instance, the ongoing DESTINY-Breast09 trial is evaluating the efficacy of trastuzumab deruxtecan with or without trastuzumab, compared to chemotherapy combined with trastuzumab and pertuzumab for first-line treatment of HER2-positive MBC. In addition, the HER2CLIMB-05 trial aims to evaluate the benefits of adding tucatinib to chemotherapy plus trastuzumab and pertuzumab in the same setting. However, the results from these trials are still pending. Moreover, emerging as a cooperative therapeutic strategy, the combination of ADC and TKI is under active examination in HER2-positive breast cancer management. Specific ongoing trials include second-line treatment combinations such as tucatinib + T-DM1 vs. placebo + T-DM1 (NCT03975647), tucatinib + T-DXd in later lines of therapy (NCT04539938), and SHR-A1811 (an HER-2 ADC) + pyrotinib (NCT05353361), as well as adjuvant therapy with tucatinib + T-DM1 (NCT04457596). As we venture into the future, this innovative ADC + TKI model may soon permeate first-line therapeutic options. The interplay between TKIs and both large molecule monoclonal antibodies and small molecule ADCs signifies an imperative area for continued exploration, where the selection of a robust TKI could be crucial. Within this context, our study has elucidated that pyrotinib might be a viable option for patients with HER2-positive MBC. In light of your feedback, we have updated the introduction and discussion of our manuscript to reflect this information, thus providing a more comprehensive overview of the current and evolving therapeutic landscape for HER2-positive MBC.

Here is added in the introduction:

The treatment paradigm for HER2-positive MBC is continually evolving, with new developments such as ADCs like trastuzumab deruxtecan [13, 14] and TKIs including tucatinib [15, 16]. For example, the ongoing DESTINY-Breast09 trial is assessing the impact of trastuzumab deruxtecan, with or without trastuzumab, versus the combination of chemotherapy, trastuzumab, and pertuzumab as a first-line treatment for HER2-positive MBC (NCT04784715). Furthermore, the HER2CLIMB-05 trial is exploring the potential benefits of supplementing the regimen of chemotherapy, trastuzumab, and pertuzumab with tucatinib (NCT05132582). Nevertheless, the outcomes from these trials are yet to be reported.

Here is added in the discussion:

The therapeutic landscape for HER2-positive MBC is undergoing significant transformation, propelled by the advent of advanced agents such as ADCs, exemplified by trastuzumab deruxtecan [13, 14], and TKIs. The DESTINY-Breast09 trial (NCT04784715) is examining trastuzumab deruxtecan, either alone or alongside trastuzumab, against a conventional frontline regimen of chemotherapy, trastuzumab, and pertuzumab for HER2-positive MBC. Concurrently, the HER2CLIMB-05 trial (NCT05132582) is exploring the addition of tucatinib to the standard chemotherapy-trastuzumab-pertuzumab combination. The results of these investigative endeavors, however, remain pending. Moreover, there's burgeoning interest in synergizing ADCs and TKIs for therapeutic amplification. This is underscored by several ongoing studies targeting HER2-positive breast cancer: tucatinib plus T-DM1 in second-line and adjuvant settings (NCT03975647, NCT04457596); tucatinib plus trastuzumab deruxtecan in later-line settings (NCT04539938); SHR-A1811, a HER-2 specific ADC, plus pyrotinib, in a second-line setting (NCT05353361). As we venture into the future, this innovative ADC plus TKI model may soon permeate first-line therapeutic options. The interplay between TKIs and both large molecule monoclonal antibodies and small molecule ADCs signifies an imperative area for continued exploration, where the selection of a robust TKI could be crucial. Within this context, our study has elucidated that pyrotinib might be a viable option for patients with HER2-positive MBC.

Finally, the results reported here concern a small group of patients, included in a phase II trial with a new molecule, whose development is limited to a certain part of the world only, and which, despite encouraging results, does not seem to bring a significant benefit compared to current standards, and to new strategies in development such as ADCs targeting HER2.

Response: We appreciate your thoughtful review and recognition of the challenges that come with conducting a phase II trial with a novel molecule and in a specific geographical context, which has been mentioned in our Discussion (Firstly, it is a single-arm study with a limited sample size, which may introduce potential bias and limit the generalizability of the findings. Secondly, the study only enrolled Chinese patients, and the efficacy and safety of pyrotinib plus docetaxel in other populations remain unknown. Further research in diverse populations is needed to validate the results). As you correctly pointed out, our study involves the use of pyrotinib in the first-line treatment setting where an earlier anti-HER2 TKI, lapatinib, failed to show positive results. The

unmet need in this context led us to investigate the potential of this new molecule, and we believe that our preliminary results serve to fill this gap, despite the inherent limitations. Furthermore, it is crucial to note that while advanced therapies like ADCs and TKIs are indeed promising, the results from first-line treatment trials, such as DESTINY-Breast09 and HER2CLIMB-05, are still pending. Moreover, emerging as a cooperative therapeutic strategy, the combination of ADC and TKI is under active examination in HER2-positive breast cancer management. Specific ongoing trials include second-line treatment combinations such as tucatinib + T-DM1 vs. placebo + T-DM1 (NCT03975647), tucatinib + T-DXd in later lines of therapy (NCT04539938), and SHR-A1811 (an HER-2 ADC) + pyrotinib (NCT05353361), as well as adjuvant therapy with tucatinib + T-DM1 (NCT04457596). As we venture into the future, this innovative ADC + TKI model may soon permeate first-line therapeutic options. The interplay between TKIs and both large molecule monoclonal antibodies and small molecule ADCs signifies an imperative area for continued exploration, where the selection of a robust TKI could be crucial. Within this context, our study has elucidated that pyrotinib might be a viable option for patients with HER2-positive MBC. Lastly, our study still provides preliminary safety and efficacy data for the phase III study PHILA. Once again, we thank you for your insightful comments and aim to continue contributing to this crucial area of research.

Here is added in the discussion:

The therapeutic landscape for HER2-positive MBC is undergoing significant transformation, propelled by the advent of advanced agents such as ADCs, exemplified by trastuzumab deruxtecan [13, 14], and TKIs. The DESTINY-Breast09 trial (NCT04784715) is examining trastuzumab deruxtecan, either alone or alongside trastuzumab, against a conventional frontline regimen of chemotherapy, trastuzumab, and pertuzumab for HER2-positive MBC. Concurrently, the HER2CLIMB-05 trial (NCT05132582) is exploring the addition of tucatinib to the standard chemotherapy-trastuzumab-pertuzumab combination. The results of these investigative endeavors, however, remain pending. Moreover, there's burgeoning interest in synergizing ADCs and TKIs for therapeutic amplification. This is underscored by several ongoing studies targeting HER2-positive breast cancer: tucatinib plus T-DM1 in second-line and adjuvant settings (NCT03975647, NCT04457596); tucatinib plus trastuzumab deruxtecan in later-line settings (NCT04539938); SHR-

A1811, a HER-2 specific ADC, plus pyrotinib, in a second-line setting (NCT05353361). As we venture into the future, this innovative ADC plus TKI model may soon permeate first-line therapeutic options. The interplay between TKIs and both large molecule monoclonal antibodies and small molecule ADCs signifies an imperative area for continued exploration, where the selection of a robust TKI could be crucial. Within this context, our study has elucidated that pyrotinib might be a viable option for patients with HER2-positive MBC.

On the other hand, the lack of new translational biological results makes me think that this article should be submitted to a clinical oncology journal and not to a journal such as Nature Communication which does not seem to be adapted to the scientific, biological and innovative content of this publication.

Response: Thank you very much for your thoughtful feedback and suggestions. We genuinely appreciate your point of view on the suitability of our manuscript for Nature Communications. We do agree that our research, at its core, is clinical in nature. However, we also believe it embodies a translational aspect that aligns with the diverse scope of Nature Communications. Specifically, our identification of PIK3CA mutations and others as potential biomarkers of treatment outcome echoes with previous studies, and adds to the ongoing scientific dialogue on personalized therapies for HER2-positive MBC. Furthermore, we see the multidisciplinary aspect of Nature Communications as a strength and an opportunity. The journal's broad readership across various scientific disciplines aligns with our intention to disseminate our findings widely. Lastly, please allow us to express our respect and appreciation for the high standards that Nature Communications upholds. It is precisely for these high standards that we chose to submit our research to this journal. We believe that our study, despite its limitations, represents a potential valuable step forward in the search for improved treatment strategies for HER2-positive MBC. We hope you would kindly reconsider the merits of our work in light of these explanations, and we look forward to any further feedback you might have.

Reviewer #3

This is a phase study aimed to investigate the efficacy and safety of pyrotinib plus docetaxel as first-line treatment in HER2-positive MBC. The confirmed ORR was 79.7% (95% confidence interval [CI], 70.8-88.6) in the intention-to-treat population. The median PFS 16 was 16.0 months (95% CI, 11.2-20.8). This trial is of interest, since more than 30% of patients had received trastuzumab in their (neo)adjuvant setting in this study. However, there are several aspects that require further explanation:

1) This is a single-arm study with a limited sample size, authors should avoid comparing the efficacy results observed here with the CLEOPATRA study and other phase 3 randomized controlled trials.

Response: Thank you for your constructive comment. We fully recognize the limitations inherent in our study design, specifically being a single-arm study with a restricted sample size. We have noted this in the limitations section of our discussion (Firstly, it is a single-arm study with a limited sample size, which may introduce potential bias and limit the generalizability of the findings). Upon your suggestion, we have revisited the discussion section and toned down our language to avoid direct comparisons with larger, randomized controlled trials such as the CLEOPATRA study. We acknowledge that while our findings may add potential value to the overall research landscape, it is not wholly accurate or fair to make direct comparisons to those more robust trials. We aimed to present our findings in the context of the current research landscape rather than suggesting a comparison of efficacy.

2) Nearly 70% of patients in the study had not received trastuzumab in their (neo)adjuvant setting, I am concerning about using a small molecule TKI plus docetaxel other than the standard THP for these patients. Moreover, I think it will be more interesting to explore the efficacy and safety of pyrotinib in combination with Herceptin and docetaxel.

Response: We greatly appreciate your valuable insights. Indeed, as you rightly point out, the current first-line standard treatment for HER2-positive MBC is a combination of chemotherapy with trastuzumab and pertuzumab, based on the results of the CLEOPATRA trial [2-4]. Nevertheless, the role of anti-HER2 TKIs as part of the first-line treatment regimen for HER2-positive MBC remains an area of ongoing debate. For comparison, the lapatinib plus taxanes regimen, another TKI-based treatment, showed a shorter median PFS (9.0 months) and raised more safety concerns in treatment-

naive HER2-positive MBC patients than trastuzumab plus taxanes [5]. Despite this, prior research demonstrated the superiority of pyrotinib over lapatinib when combined with capecitabine in patients previously treated for HER2-positive MBC [1, 6]. Before our study, however, the role of pyrotinib combined with chemotherapy as a first-line treatment for HER2-positive MBC had not been explored. We concur with your observation that the use of pyrotinib in combination with trastuzumab and docetaxel would be intriguing. In fact, the insights gleaned from our study have paved the way for the phase III PHILA trial, which was initiated after our research had begun. The PHILA trial demonstrated that the combination of pyrotinib, trastuzumab, and docetaxel significantly prolongs PFS compared to placebo plus trastuzumab and docetaxel in the first-line treatment of HER2-positive MBC patients [7]. In essence, our study offers fresh insights on the potential of pyrotinib plus docetaxel as a first-line treatment regimen and supplies crucial preliminary data for the phase III PHILA trial. Our hope is that our findings, in conjunction with ongoing research, will aid the evolution of new and effective treatment strategies for patients with HER2-positive MBC. We are grateful for your thorough review and invaluable input, which have helped us better articulate the unique contributions of our research.

Following is revised in our Discussion:

Dual-targeted therapy has become the standard first-line treatment for patients with MBC [8]. The CLEOPATRA study provided evidence supporting the superiority of pertuzumab plus trastuzumab and docetaxel over trastuzumab plus docetaxel in terms of PFS and OS [9]. The Chinese bridging PUFFIN study also confirmed the benefits of dual-targeted therapy in untreated HER2-positive MBC [10, 11]. However, the use of anti-HER2 TKIs in the first-line treatment of HER2-positive MBC has been a subject of ongoing debate. Studies comparing lapatinib plus taxanes to trastuzumab plus taxanes showed that the former resulted in shorter PFS (median PFS: 11.3 months vs. 9.0 months) and raised more safety concerns for untreated HER2-positive MBC patients [5]. Similarly, neratinib plus paclitaxel did not demonstrate superiority over trastuzumab plus paclitaxel in first-line treatment, with both arms showing a median PFS of 12.9 months [12]. Prior research highlighted the superiority of pyrotinib over lapatinib when combined with capecitabine in patients previously treated for HER2-positive MBC [1, 6]. However, the efficacy and safety of pyrotinib plus chemotherapy as a first-line treatment for HER2-positive MBC remained unexplored before

our study. Our investigation aimed to address this knowledge gap and shed light on the potential role of pyrotinib plus chemotherapy in the first-line treatment of HER2-positive MBC. In our study, the combination of pyrotinib and docetaxel resulted in a confirmed ORR of 79.7% and a median PFS of 16.0 months. Furthermore, the treatment demonstrated rapid and durable responses, with a median TTR of 1.5 months and a median DOR of 15.9 months. These findings provided preliminary efficacy and safety data that paved the way for the initiation of the phase III PHILA study. Notably, the phase III PHILA study confirmed that pyrotinib plus trastuzumab and docetaxel significantly prolonged PFS compared to placebo plus trastuzumab and docetaxel (median PFS: 24.3 months vs. 10.4 months) in the first-line treatment of HER2-positive MBC patients [7]. These results further demonstrate the potential benefits of pyrotinib in the first-line setting for HER2-positive MBC patients.

3) Prior studies using the small molecule TKI pyrotinib showed high percentages of diarrhea, and I have noticed that this study mentioned a modification of study protocol due to a high incidence of diarrhea, leukopenia, and/or neutropenia during the first stage of the study. Thus, this AE also limited the use of pyrotinib in the first-line treatment.

Response: We appreciate your observation regarding the incidence of AEs in our study. As you pointed out, leukopenia and neutropenia are typically associated with chemotherapy, and diarrhea has been a common AE observed with the use of small molecule TKIs such as lapatinib, pyrotinib, and neratinib. It's worth noting that the occurrence of severe diarrhea usually happens within the first week to one month of starting the medication, most commonly between days 1-10, with a median onset time of 2-5 days according to previous studies [5, 6, 12, 17]. Recognizing these patterns has allowed us to explore preemptive measures to manage these AEs more effectively. Indeed, the CONTROL study demonstrated that proactive management strategies, including primary prophylaxis or dose escalation, improved the tolerability of neratinib, reduced the incidence and severity of diarrhea, and decreased treatment discontinuations due to diarrhea [18]. In our study, we applied a similar approach, using loperamide as primary prophylaxis to manage potential diarrhea, leading to significantly lower incidence rates of grade ≥ 3 diarrhea in patients who received prophylaxis (8.9%; 4/45) compared to those who did not (38.2%; 13/34). Additionally, a smaller proportion of patients receiving loperamide prophylaxis experienced treatment discontinuation

(6.7%, 3/45) and dose reduction (17.8%, 8/45) in contrast to those who did not receive the prophylaxis, where the rates were 14.7% (5/34) and 35.3% (12/34), respectively. In conclusion, while the AEs you noted are indeed challenges when using TKIs like pyrotinib, our study suggests that with proactive and effective management, such as primary prophylaxis, we can mitigate these AEs and enable patients to continue their treatment. We believe this presents a promising avenue to optimize the therapeutic benefits of TKIs in the first-line treatment of HER2-positive MBC. Thank you for your insightful comments, which have allowed us to further clarify this aspect of our study.

Following is revised in our Discussion:

Diarrhea has been a common AE associated with the use of small molecule anti-HER2 TKIs such as lapatinib, pyrotinib, and neratinib [5, 6, 12, 17]. The development of diarrhea can significantly impact the quality of life for patients and may even lead to dose reductions or treatment interruptions, which can compromise the effectiveness of the treatment [19]. The CONTROL study demonstrated that proactive management strategies, such as primary prophylaxis or dose escalation, can enhance the tolerability of neratinib, reduce the incidence and severity of diarrhea, and decrease treatment discontinuations due to diarrhea [18]. In our study, we implemented loperamide as primary prophylaxis to manage potential diarrhea, resulting in significantly lower incidence rates of grade ≥ 3 diarrhea in patients who received prophylaxis (8.9%; 4/45) compared to those who did not (38.2%; 13/34). Furthermore, patients who received loperamide prophylaxis had a lower incidence of treatment discontinuation (6.7%, 3/45) and dose reduction (17.8%, 8/45) compared to those without prophylaxis, who had rates of 14.7% (5/34) and 35.3% (12/34) respectively.

4) The introduction section lack important information to contextualize the current trial.

Response: We appreciate your comments and recognize the exciting progress being made in this area with the development of new therapeutic agents. The landscape of HER2-positive MBC treatment is indeed evolving, with the emergence of ADCs such as trastuzumab deruxtecan and other TKIs such as tucatinib. Trastuzumab deruxtecan have shown great promise due to their ability to deliver cytotoxic drugs directly to HER2-positive cancer cells, thereby maximizing efficacy and minimizing systemic toxicity [13, 14]. Similarly, tucatinib has shown outstanding results, particularly in patients with brain metastases, an area where many HER2-targeted therapies struggle

[15, 16]. These agents are currently undergoing significant development, with ongoing trials evaluating their efficacy and safety in first-line settings for HER2-positive metastatic breast cancer. For instance, the ongoing DESTINY-Breast09 trial is evaluating the efficacy of trastuzumab deruxtecan with or without trastuzumab, compared to chemotherapy combined with trastuzumab and pertuzumab for first-line treatment of HER2-positive MBC. In addition, the HER2CLIMB-05 trial aims to evaluate the benefits of adding tucatinib to chemotherapy plus trastuzumab and pertuzumab in the same setting. However, the results from these trials are still pending. Moreover, emerging as a cooperative therapeutic strategy, the combination of ADC and TKI is under active examination in HER2-positive breast cancer management. Specific ongoing trials include second-line treatment combinations such as tucatinib + T-DM1 vs. placebo + T-DM1 (NCT03975647), tucatinib + T-DXd in later lines of therapy (NCT04539938), and SHR-A1811 (an HER-2 ADC) + pyrotinib (NCT05353361), as well as adjuvant therapy with tucatinib + T-DM1 (NCT04457596). As we venture into the future, this innovative ADC + TKI model may soon permeate first-line therapeutic options. The interplay between TKIs and both large molecule monoclonal antibodies and small molecule ADCs signifies an imperative area for continued exploration, where the selection of a robust TKI could be crucial. Within this context, our study has elucidated that pyrotinib might be a viable option for patients with HER2-positive MBC. In light of your feedback, we have updated the introduction and discussion of our manuscript to reflect this information, thus providing a more comprehensive overview of the current and evolving therapeutic landscape for HER2-positive MBC.

Here is added in the introduction:

The treatment paradigm for HER2-positive MBC is continually evolving, with new developments such as ADCs like trastuzumab deruxtecan [13, 14] and TKIs including tucatinib [15, 16]. For example, the ongoing DESTINY-Breast09 trial is assessing the impact of trastuzumab deruxtecan, with or without trastuzumab, versus the combination of chemotherapy, trastuzumab, and pertuzumab as a first-line treatment for HER2-positive MBC (NCT04784715). Furthermore, the HER2CLIMB-05 trial is exploring the potential benefits of supplementing the regimen of chemotherapy, trastuzumab, and pertuzumab with tucatinib (NCT05132582). Nevertheless, the outcomes from these trials are yet to be reported.

Here is added in the discussion:

The therapeutic landscape for HER2-positive MBC is undergoing significant transformation, propelled by the advent of advanced agents such as ADCs, exemplified by trastuzumab deruxtecan [13, 14], and TKIs. The DESTINY-Breast09 trial (NCT04784715) is examining trastuzumab deruxtecan, either alone or alongside trastuzumab, against a conventional frontline regimen of chemotherapy, trastuzumab, and pertuzumab for HER2-positive MBC. Concurrently, the HER2CLIMB-05 trial (NCT05132582) is exploring the addition of tucatinib to the standard chemotherapy-trastuzumab-pertuzumab combination. The results of these investigative endeavors, however, remain pending. Moreover, there's burgeoning interest in synergizing ADCs and TKIs for therapeutic amplification. This is underscored by several ongoing studies targeting HER2-positive breast cancer: tucatinib plus T-DM1 in second-line and adjuvant settings (NCT03975647, NCT04457596); tucatinib plus trastuzumab deruxtecan in later-line settings (NCT04539938); SHR-A1811, a HER-2 specific ADC, plus pyrotinib, in a second-line setting (NCT05353361). As we venture into the future, this innovative ADC plus TKI model may soon permeate first-line therapeutic options. The interplay between TKIs and both large molecule monoclonal antibodies and small molecule ADCs signifies an imperative area for continued exploration, where the selection of a robust TKI could be crucial. Within this context, our study has elucidated that pyrotinib might be a viable option for patients with HER2-positive MBC.

5) 77.2% of the patients enrolled has a ECOG of 1. This is unexpected in a trial enrolling untreated HER2-positive metastatic patients. Authors should explain.

Response: We appreciate your observation regarding the high percentage of patients with an ECOG performance status of 1 in our study. One primary reason for the relatively high ECOG performance status may be related to the high proportion of visceral metastases in our patient population. In our study, 75.9% of patients presented with visceral metastases, which often result in more pronounced symptoms and can lead to a lower performance status. Besides, an ECOG score of 1, indicating that patients are restricted in physically strenuous activity but ambulatory and able to carry out light work, is not uncommon in populations of patients with metastatic disease,

particularly at the time of initial diagnosis. A significant number of these patients may be minimally symptomatic at presentation. We hope this explanation addresses your concern.

6) Authors should improve the language in this study.

Response: Thank you for your feedback regarding the language used in our manuscript. Following your suggestion, we have made significant efforts to improve the language and readability of our manuscript. We hope that these revisions have enhanced the clarity and coherence of our manuscript.

Reference:

1. Ma, F., et al., *Pyrotinib or Lapatinib Combined With Capecitabine in HER2-Positive Metastatic Breast Cancer With Prior Taxanes, Anthracyclines, and/or Trastuzumab: A Randomized, Phase II Study*. J Clin Oncol, 2019. **37**(29): p. 2610-2619.
2. Baselga, J., et al., *Pertuzumab plus trastuzumab plus docetaxel for metastatic breast cancer*. N Engl J Med, 2012. **366**(2): p. 109-19.
3. Swain, S.M., et al., *Pertuzumab, trastuzumab, and docetaxel in HER2-positive metastatic breast cancer*. N Engl J Med, 2015. **372**(8): p. 724-34.
4. Swain, S.M., et al., *Pertuzumab, trastuzumab, and docetaxel for HER2-positive metastatic breast cancer (CLEOPATRA): end-of-study results from a double-blind, randomised, placebo-controlled, phase 3 study*. Lancet Oncol, 2020. **21**(4): p. 519-530.
5. Gelmon, K.A., et al., *Lapatinib or Trastuzumab Plus Taxane Therapy for Human Epidermal Growth Factor Receptor 2-Positive Advanced Breast Cancer: Final Results of NCIC CTG MA.31*. J Clin Oncol, 2015. **33**(14): p. 1574-83.
6. Xu, B., et al., *Pyrotinib plus capecitabine versus lapatinib plus capecitabine for the treatment of HER2-positive metastatic breast cancer (PHOEBE): a multicentre, open-label, randomised, controlled, phase 3 trial*. Lancet Oncol, 2021. **22**(3): p. 351-360.
7. Xu, B., et al., *LBA19 Pyrotinib or placebo in combination with trastuzumab and docetaxel for HER2-positive metastatic breast cancer (PHILA): A randomized phase III trial*. Annals of Oncology, 2022. **33**: p. S1387.
8. Loibl, S., et al., *Breast cancer*. Lancet, 2021. **397**(10286): p. 1750-1769.
9. Swain, S.M., et al., *Pertuzumab, trastuzumab, and docetaxel for HER2-positive metastatic breast cancer (CLEOPATRA study): overall survival results from a randomised, double-blind, placebo-controlled, phase 3 study*. Lancet Oncol, 2013. **14**(6): p. 461-71.
10. Xu, B., et al., *Pertuzumab, trastuzumab, and docetaxel for Chinese patients with previously untreated HER2-positive locally recurrent or metastatic breast cancer (PUFFIN): a phase III, randomized, double-blind, placebo-controlled study*. Breast Cancer Res Treat, 2020. **182**(3): p. 689-697.
11. Xu, B., et al., *196P Final analysis of PUFFIN: A phase III, randomised double-blind, placebo (Pla)-controlled study of pertuzumab, trastuzumab and docetaxel (PHD) for Chinese patients (pts) with previously untreated HER2-positive locally recurrent or metastatic breast cancer (LR/MBC)*. Annals of Oncology, 2022. **33**: p. S217-S218.
12. Awada, A., et al., *Neratinib Plus Paclitaxel vs Trastuzumab Plus Paclitaxel in Previously Untreated Metastatic ERBB2-Positive Breast Cancer: The NEfERT-T Randomized Clinical Trial*. JAMA Oncol, 2016. **2**(12): p. 1557-1564.
13. Modi, S., et al., *Trastuzumab Deruxtecan in Previously Treated HER2-Positive Breast Cancer*. N Engl J Med, 2020. **382**(7): p. 610-621.
14. Cortes, J., et al., *Trastuzumab Deruxtecan versus Trastuzumab Emtansine for Breast Cancer*. N Engl J Med, 2022. **386**(12): p. 1143-1154.
15. Murthy, R.K., et al., *Tucatinib, Trastuzumab, and Capecitabine for HER2-Positive Metastatic Breast Cancer*. N Engl J Med, 2020. **382**(7): p. 597-609.
16. Curigliano, G., et al., *Tucatinib versus placebo added to trastuzumab and capecitabine for patients with pretreated HER2+ metastatic breast cancer with and without brain metastases (HER2CLIMB): final overall survival analysis*. Ann Oncol, 2022. **33**(3): p. 321-329.

17. Yan, M., et al., *Pyrotinib plus capecitabine for human epidermal factor receptor 2-positive metastatic breast cancer after trastuzumab and taxanes (PHENIX): a randomized, double-blind, placebo-controlled phase 3 study*. Translational Breast Cancer Research, 2020. **1**.
18. Barcenas, C.H., et al., *Improved tolerability of neratinib in patients with HER2-positive early-stage breast cancer: the CONTROL trial*. Ann Oncol, 2020. **31**(9): p. 1223-1230.
19. Li, J., *Diarrhea With HER2-Targeted Agents in Cancer Patients: A Systematic Review and Meta-Analysis*. J Clin Pharmacol, 2019. **59**(7): p. 935-946.

REVIEWERS' COMMENTS

Reviewer #1 (Remarks to the Author):

Most statistical issues have been addressed. Two minor comments for consideration.

1. Questionable definition of duration of response (DoR), defined as the time from the first documented response to disease progression for patients with confirmed CR and PR, and the time from treatment initiation to disease progression for patients with SD. Since it includes patients with SD, duration of disease control is more appropriate.
2. Recommend removing genes with $n < 4$ in Table 4 since statistical significance becomes meaningless for $n = 1-2$ (e.g., questionable HR = 3931326707745.28).

Reviewer #2 (Remarks to the Author):

I sincerely appreciate the care with which the authors have responded to my comments point by point.

I maintain the reservations I initially expressed concerning :

- The absence of overall survival data
- The fact that these results do not change the current management algorithm for HER2+ metastatic breast cancer.
- Current and future development of pyrotinib seems limited to certain geographic markets, and constrained by the current development of other TKIs (tucatinib) and highly effective new-generation ADCs.
- The very limited translational biology data provided with this article.

However, the authors have discussed each of these limitations in the revised version of the manuscript, which adds nuance to the data presented.

Reviewer #3 (Remarks to the Author):

The author has adequately addressed all the reviewer's concerns in the current revised version of the manuscript. Therefore, the reviewer suggests acceptance of the manuscript in its current form.

Reviewer #1 (Remarks to the Author):

Most statistical issues have been addressed. Two minor comments for consideration.

1. *Questionable definition of duration of response (DoR), defined as the time from the first documented response to disease progression for patients with confirmed CR and PR, and the time from treatment initiation to disease progression for patients with SD. Since it includes patients with SD, duration of disease control is more appropriate.*

Response: Thank you for highlighting the concern regarding our definition of DoR. Upon reevaluation, we realize our oversight in the description. After a thorough review of our data and consultation with our statistician, we confirm that in our study, DoR was indeed defined as the time from the first documented response to disease progression specifically for patients with confirmed CR and PR, and it did not include patients with SD. Hence, we believe that retaining the term “DoR” in our manuscript is appropriate. We apologize for any confusion caused.

2. *Recommend removing genes with $n < 4$ in Table 4 since statistical significance becomes meaningless for $n = 1-2$ (e.g., questionable $HR = 3931326707745.28$).*

Response: Thank you for highlighting the concern regarding genes with a low sample size in Table 4. We understand that small sample sizes can lead to questionable statistical significance. In response to this feedback, we have removed genes with $n < 4$ from Table 4 to ensure the reliability of our presented data.

Reviewer #2 (Remarks to the Author):

I sincerely appreciate the care with which the authors have responded to my comments point by point.

I maintain the reservations I initially expressed concerning :

- The absence of overall survival data*
- The fact that these results do not change the current management algorithm for HER2+ metastatic breast cancer.*
- Current and future development of pyrotinib seems limited to certain geographic markets, and constrained by the current development of other TKIs (tucatinib) and highly effective new-generation ADCs.*

- The very limited translational biology data provided with this article.

However, the authors have discussed each of these limitations in the revised version of the manuscript, which adds nuance to the data presented.

Response: We genuinely appreciate the time and effort you've dedicated to reviewing our manuscript. Your feedback has not only improved the present manuscript but will also guide our future endeavors. Thank you once again.

Reviewer #3 (Remarks to the Author):

The author has adequately addressed all the reviewer's concerns in the current revised version of the manuscript. Therefore, the reviewer suggests acceptance of the manuscript in its current form.

Response: Thank you for taking the time to review our work and for your constructive feedback. We are grateful for your positive assessment and recommendation for acceptance.